

# Unraveling the hydrological budget of isolated and seasonally contrasted sub-tropical lakes

Chloé Poulin[1], Bruno Hamelin[1], Christine Vallet-Coulomb[1], Guinbe Amngar[3], Bichara Loukman[3], Jean-François Cretaux[4], Jean-Claude Doumnang[3], Abdallah Mahamat Nour[1,3], Guillemette Menot[1,2], Florence Sylvestre[1], and Pierre Deschamps[1]

[1]Aix-Marseille Université, CNRS, IRD, Collège de France, CEREGE, Europole de l Arbois 13545 Aix-en-Provence, France
[2]Univ. Lyon, Ens de Lyon, Université Lyon 1, CNRS, UMR 5276 LGL-TPE, 69342 Lyon, France
[3]Département de Géologie, Faculté des sciences Exactes et Appliquées, Université de NDjaména, NDjaména, Tchad
[4]Legos, UMR5566, 14 Avenue Edouard Belin, 31400 Toulouse, France

**Correspondence:** Chloé Poulin(poulin@cerege.fr)

**Abstract.** Complete understanding of the hydrological functioning of large scale intertropical watersheds like the Lake Chad basin, which become a high priority in the prospect of near future climate change and increasing demographic pressure, require integrated studies of all surface and groundwater reservoirs and their quite complex interconnections. In this respect, detailed hydrological studies of secondary peripheral lakes of these large basins may provide us with interesting small scale analogs of the major waterbodies, which can help disentangling the multiple influences of various forcing factors of the water cycle and its evolution.

We present here a simple method for estimating the annual mean water balance of sub-sahelian lakes subject to high seasonal contrast, and located in isolated regions with no road access during the rain season, precluding continuous monitoring of in-situ hydrological data.

The approach is illustrated by the study of the two lakes Iro and Fitri in the eastern basin of lake Chad, so far unstudied, and also tested on lake Ihotry (Madagascar), extensively studied previously by our group.

We combine the isotopic data ($\delta^{18}O; \delta^2H$) that we measured during the dry season with altimetry data from the SARAL satellite mission, in order to model the seasonal variation of lake volume and isotopic composition. The annual water budget is then estimated from mass balance equations using the Craig and Gordon's model for evaporation. We show that the closed-system behavior (precipitation equal to evaporation) can be confirmed for lake Ihotry, whereas we calculate E/I ratios of $0.6 \pm 0.3$ and $0.4 \pm 0.2$ for Iro and Fitri, respectively, in both cases compatible with water fluxes estimated from nearby gauging stations. In the case of Fitri the estimated output flux is contributing to the groundwater recharge, since the lake has no identified surface outlet.

Finally, we use our data to discuss possible inferences about the hydro-climatic budget of the catchment basins of those two lakes. We show that the average rainfall isotopic composition monitored by IAEA at NDjamena is slightly offset from the two distinct Local Evaporation Lines that we obtain on the two lake-aquifer systems, and that this slight difference may reflect the impact of vegetation transpiration on the basin water budget. We conclude that, while being broadly consistent with

(c) Author(s) 2018. CC BY 4.0 License.





transpiration being on the same order of magnitude as evaporation in those basins, we cannot derive a more precise estimate of the partition between these two fluxes, owing to the large uncertainties of the different end-members in the budget equations.

*Copyright statement.* TEXT

# 1 Introduction

In Sahel, the combined effects of population growth, land degradation and changes in rainfall patterns already resulted in a significant deterioration of soil and water resources (UNEP and ICRAF, 2006). Since the 19th century, the Sahel region suffered from successive droughts and wet periods (Nicholson, 2013) : following the severe droughts that ravaged the region in the 1970s and 1980s, an increase in rainfall pattern has been observed in the central and western parts since the 1990s (Nicholson, 2005; Lebel and Ali, 2009; Ali and Lebel, 2009). So far, there is no consensus on precipitation trends for the 21st

century (Druyan, 2011; Defrance et al., 2017), leading to large uncertainties on the evolution of the Sahelian lake systems. In sub-saharian areas, most of these lakes are endorheic systems in which surface runoff accumulates during the rainy season, forming reservoirs for agriculture and human activities. Their vulnerability to changes in climatic conditions make thus the understanding of their hydrological functioning essential to socio-economic previsions. Most studies published so far on these systems face a lack of long-term hydrological monitoring of the watersheds (Sivapalan et al., 2003). Remote sensing techniques

have been developed to compensate for this lack of data, (Liebe et al., 2005; Rodrigues et al., 2012; Gal et al., 2016) in order to determine lake storage capacities or to estimate inflows. In parallel, geochemical and isotopic data can provide independent quantitative constraints to determine lakes dynamic or steady state water balance. Since the pioneer investigations in this domain (Friedman et al., 1964; Dinçer, 1968; Gat and Gonfiantini, 1981) stable isotopes were used extensively, with specific insights in particular on surface-groundwater interactions (Krabbenhoft et al., 1990; Sacks et al., 2014; Bouchez et al., 2016)

and evaporative flux characterization (Gibson et al., 2002; Mayr et al., 2007; Gibson et al., 2017).
Studies of inter-tropical lakes raise specific difficulties, mainly related to intrinsic characteristics such as the extremely high evaporation rates, added to high transpiration of aquatic vegetation, huge seasonal variations of fluxes and lake level, and seasonal changes in hydrologic configuration resulting from ephemeral swamps and humid zones flooded during the wet season. Moreover, an even more compelling problem stems from the logistical impossibility to reach the field sites during the wet

season, when the tracks network is virtually impracticable during several months. As a consequence, we are generally missing critical data for the recharge period, to infer seasonal ranges of variation, and thus annual average values.

In this study, we present an approach to circumvent these difficulties, based on the combination of dry season measurements of O and H stable isotopes in the lake and surrounding groundwater, with remote sensing monitoring of the seasonal lake level

variations using Satellite Altimetry. We test this approach on the two peripheral lakes Iro and Fitri included in Lake Chad basin. Understanding the hydrology of Lake Chad, and the origin of its strong surface variability, has been focus of a large number of





studies (Fontes et al., 1970; Carmouze, 1969; Olivry et al., 1996; Bouchez et al., 2016). However predicting its future behavior in response to climate change still remains challenging owing to the complexities of its hydrography, and extremely diverse characteristics of the various compartments of its catchment. We show here that the study of secondary lakes of the same basin, besides their importance for local populations, may provide us with pertinent small scale analogs of Lake Chad itself, and help

testing the respective influence of the different forcing parameters and processes, thus constituting potential sentinel systems for future evolutions. In a first part we will show how to combine isotopic data with satellite data to obtain a first annual water balance at lake scale. Then we will see how transpiration and evaporation can affect the isotopic signal and the water balance at catchment scale.

## 2 Study area

Lakes Iro (10.1° N ; 19.4° E) and Fitri (12.8° N ; 17.5° E) belong to two different sub-catchments in the central part of the lake Chad Basin (Fig.1). Iro is close to the outlet of the Bahr-Salamat watershed (195 000 $km^2$), a sub-catchment of the Chari-Logone (90 % of the Lake Chad water supply). Bahr-Salamat takes its source in Darfour (North Sudan) and feeds Lake Iro during the rainy season through a seasonal defluent, Fig.2 (Billon et al., 1974). Lake Fitri (Fig.3) is the endorheic terminal lake of the Batha catchment (96 000 km2), which lies entirely in the Sahel zone. These two lakes are thus located under two

different climates: sahelo-sudanese for Iro, and sahelian for Fitri. They are both characterized by a rainy season between June and September, followed by a dry season, with an annual rainfall of 765 $mm\,yr^{-1}$ at the Am Timan station upstream Iro, and 360 $mm\,yr^{-1}$ at Ati station upstream Fitri (annual averages between 1960 and 2014, Direction of water resources and meteorology: DREM, Chad). Precipitation are brought by the African Monsoon, linked to the northward shift of the Intertropical Convergence Zone (ITCZ), and characterized by a large inter-annual as well as inter-decadal variability: the 50's and 60's

received heavy precipitation, while the 70's and 80's recorded a severe drought. The mean annual temperatures are 27°C and 28°C for Iro and Fitri respectively, and their relative mean annual humidity is 50 % and 40 % (data from Am Timan station between 1966 and 1976 for Iro, and from Ati station between 1960 and 2004 for Fitri (Boyer et al., 2006)). Piche evaporimeter data were recorded between 1961 and 1988 at the Birao station (same latitude as Iro but further east) in Central African Republic, giving a 1.8 $m\,yr^{-1}$ evaporative flux (DREM).

Fluvial flow rates were measured on the Bahr Azoum, upstream of Iro in the Bahr Salamat catchment (ORSTOM data from 1953 to 1966, then DREM until 1973), and upstream of Fitri in the Batha catchment, between 1955 and 1993 (SIEREM data, (Boyer et al., 2006)).

The rivers (also locally called "Bahrs") flow on the Quaternary deposits of the Chad basin. A few granitic inselbergs outcrop around the two lakes, along with laterite surfaces especially at Iro. The superficial aquifers are imbedded in sand-clays sed-

imentary deposits. The aquifer is uniformly shallow (5-30 m) around Iro, while the depth increases around Fitri, from the riverbank (10-15 m in Ati) toward the North, down to 50 m at 20 km from the lake (Schneider, 2004).

The two lakes are surrounded mainly by hydromorphic or halomorphic vertisols (soil map ORSTOM, 1968). Hydromorphic soils are constantly flooded during the rainy season (Gillet, 1969) which allow flood recession crops (especially sorghum,



locally called "berbere"). Vegetation forms acacia-dominated savannah (*Acacia seyal* and *Acacia sieberiana*) and combretum sudan savannah (*Combretum glutinosum* and *Terminalia avicennioides*) around Iro (Gillet, 1969), and a steppe with Acacia Senegal and Acacia tortilis at Fitri.

The regions of Guera (lake Iro) and Batha (lake Fitri) each host half a million people (RGPH, 2009). These lakes form impor-
tant economic poles in their respective regions with various activities: agriculture, fishing, livestock farming, and more recently, gold mining in the south of Fitri. These multiple activities can be sources of conflict when their distribution area overlap, and also increase pressure on resources.

## 3    Materials and methods

Samples were collected for geochemical analyses during two field campaigns, on lake Fitri at the end of February 2015 and
on lake Iro in April 2015. At each sampling point (Fig.2 and 3), physico-chemical parameters were measured, and surface and groundwaters were collected for stable isotopes, major ions and $^{36}$Cl analyses. A total of 33 groundwater samples were collected (14 for lake Iro and 19 for lake Fitri) and 4 surface water (2 for each lake). The Bahr Salamat was also sampled upstream of the defluent feeding lake Iro, and one sample from the Chari river was taken upstream of the confluent of the Bahr Salamat. The Batha river was dry at this period.

Stable isotopes measurements were performed at CEREGE using cavity ring-down laser spectrometry (PICARRO L1102-i) for low salinity samples (<1000 $\mu S\,cm^{-1}$). The high salinity samples (>1000 $\mu S\,cm^{-1}$) were analysed on a dual inlet Delta Plus mass spectrometer, after equilibration with $CO_2$ (10h at 291K) and $H_2$ (2h at 291K with a platinum catalyst), for $\delta^{18}O$ and $\delta^2H$, respectively, in an automated HDO Thermo Finnigan equilibrating unit. The isotopic ratios are reported in per mil (‰) versus VSMOW, normalized to the VSMOW2-VSLAP2 scale using three laboratory standards, following the IAEA reference
sheet IAEA (2009). Each analysis has been duplicated, or more when necessary, and the total uncertainty is less than $\pm$ 0.15‰ (1$\sigma$) and $\pm$ 1‰ (1$\sigma$) for $\delta^{18}O$ and $\delta^2H$ respectively.

In order to survey the lake surface variations, we used Landsat-7 and Landsat-8 monthly satellite images (NASA Landsat Program, 1999 and 2013). Water level changes can be measured using satellite altimetry. This technique that has been designed to study the water level changes over the oceans, can also be used over lakes and rivers. Since 1992 a large number of altimetry
missions have been launched. The nadir altimeter emits pulses towards the nadir and then the measurement of the time taken to receive the echo bounced by the ground gives the distance between the altimeter and the reflecting surface. For continental hydrology, knowing the precise orbit of the satellite, this is then used to determine the water height of the lakes or the rivers (Crétaux et al., 2016). Altimetry data from AltiKa altimeter onboard the SARAL satellite was used to track the lake level variations between 2013 and 2015.



## 4   Results

Iro and Fitri share with Lake Chad itself the common characteristic of being fresh water lakes in spite of the very high evaporation rate in the Sahel region, with dry season conductivity values of 170 and 140 $\mu$S cm$^{-1}$ respectively, and a similar pH of 8. By contrast, groundwaters around the two lakes are in general more saline, although much variable, with conductivities ranging randomly between 65 and 1012 $\mu$S cm$^{-1}$ at Iro, and between 705 and 14 000 $\mu$S cm$^{-1}$ at Fitri. Measured pH are more acidic around Iro (5 < pH < 7) than around Fitri (6.5 < pH < 8.5).

The lakes stable isotopes composition are (+3.11 ‰, +12.3 ‰) for $\delta^{18}$O and $\delta^{2}$H at Iro, and (+2.04 ‰, +5.8 ‰) at Fitri. In the $\delta^{2}$H versus $\delta^{18}$O diagram (Fig.4), these compositions plot far to the right of the Global Meteoric Water Line (GMWL), with d-excess values of -12 ‰ and -10 ‰ respectively, typical of evaporated lake waters (see Jasechko et al. (2013) for a global compilation).

As classically observed in semi-arid zone, (Gaye and Edmunds, 1996; Weyhenmeyer et al., 2000; Lamontagne et al., 2005; Gonçalvès et al., 2015) the groundwater data around the two lakes form two local evaporation lines (LEL), with a slope lower than the GMWL, and the most depleted values plotting on or close to it (Fig.4). A noticeable feature is that the two lines are very close but nevertheless significantly distinct from each other, with a similar range of variation (between -5 ‰ and 0 ‰ for $\delta^{18}$O at Iro, and -5 ‰ and +2 ‰ for Fitri) and similar slopes (5.5 $\pm$ 0.3 and 5.2 $\pm$ 0.3), but a distinctly lower intercept (Fig.4). Although the two alignments are well defined, it can be noticed that the data from Fitri show more scatter than those from Iro, especially in the most depleted values (MSWD = 6.0 and 33, respectively). Finally, it must be pointed out that the lakes compositions both plot on Iro's LEL, suggesting direct connection and continuity between surface water (lake and river) and aquifer at Iro, but a more complex situation at Fitri.

The local meteoric water line (LMWL) closest to our study sites is given by the rain samples collected at the IAEA station of N'Djamena between 1962 and 2015. The data are more or less continuous between 1963 and 2015 : a total of 78 months was recorded. The slope of the alignment is 6.3 $\pm$ 0.2 with a large range of variation between -10 and +10 ‰ for $\delta^{18}$O. Such a low slope is generally considered as signature of a strong evaporation of the rain droplets during their atmospheric cycle (Dansgaard, 1964; Gat, 1996). Interestingly, the precipitation weighted average (-3.53 ‰, -18.4 ‰; d-excess=6.5‰) calculated over this period is significantly more enriched than the values found at the intersection between the local meteoric water line and the two LEL of Iro (-5.83 ‰; -36.6 ‰; d-excess=3.9‰) and Fitri (-7.21 ‰; -47.7 ‰; d-excess=-2.6‰) (Fig.4).

Figure 4 also compares our results with the data published on Lake Chad Northern pool (Fontes et al., 1970) and Southern pool (Bouchez et al., 2016). All these data lay above the LEL of Iro : data from the Northern pool are very enriched, between -0.8 ‰ and +15 ‰ for $\delta^{18}$O and between -2.8 ‰ and +77 ‰ for $\delta^{2}$H . By comparison, the Southern pool is more depleted, between -3.3 ‰ and -1.5 ‰ for $\delta^{18}$O and between -26 ‰ and -10 ‰ for $\delta^{2}$H . The isotopic composition of the Chari-Logone river (-2.55 ‰ for $\delta^{18}$O and -15.4 ‰ for $\delta^{2}$H), measured in November 2011 and between January 2013 and August 2014 (Bouchez





et al., 2016; Mahamat Nour et al., 2017), lie on the same trend as the lake Chad LEL.

Altimetry data show amplitudes between 2 and 3 meters for the lake level variations at Iro for 2013 and 2014 (year 2015 is incomplete). The measured maximum is found in August and September and the minimum in June and May for these two
years. For the same period, the amplitude was 2 meters for Fitri, with a maximum in September and a minimum in June (Fig.5). Since the water depth measured during the field campaign at Iro in April 2015 was 2 meters, uniform across the transect, these satellite observations show that the increase in water depth during the wet season was by more than a factor two. By contrast, the free water surface ($\approx 100\ \mathrm{km}^2$), clearly visible on Landsat 7 and Landsat 8 pictures (Fig.2), shows only little change during the year. This suggests that the extra 2 meters of water added to the lake in summer are in fact spread over a large area adjacent
to the lake itself, covered by vegetation tolerant to seasonal flooding. We think that the slightly lighter color observed on the satellite images around the lake (Fig.2) is likely diagnostic of this surface, that we can thus estimate on the order of 600 $\mathrm{km}^2$ (i.e. about six times larger than the dry season lake surface). The related change of volume is of a factor 6, determined by IDW interpolation and assuming a linear decrease of the depth toward the edges.

Similarly, lake Fitri also undergoes large seasonal variations of its surface: more than 1000 $\mathrm{km}^2$ wide during its high level,
it can shrink to less than 200 $\mathrm{km}^2$ at its low level (Fig.3). Based on the bathymetry measured during the second campaign (February, 2016) (2.5 meters maximum depth in the western part of the lake), the related change of volume is of a factor 4 for the year 2016.

## 5 Discussion

The discussion of our data will be divided in two separate parts. First, we analyze the implications for the determination of the
water balance at the local scale of each lake. We intend to show that a reliable estimate of the ratio between evaporation and influx (E/I) can be obtained by combining the dry season isotopic data with the satellite altimetry monitoring of the seasonal cycle. In the second part, we extrapolate the results at the regional scale of the catchment basins, trying to investigate the possible inferences of the isotopic differences between the basins, and the relationships with the rainfall data, for a better characterization of the hydroclimatic regime and inter-twinning between the controlling factors of infiltration, evaporation,
transpiration and runoff.

### 5.1 Water balance of the lakes

In this section, we first recall the basic principle of the determination of E/I and explain our handling of the isotopic budget equations. Then we test the approach on the extensively studied tropical lake Ihotry, in Madagascar. Finally, we apply the same method to the Chadian lakes Iro and Fitri, and discuss the margins of uncertainty on the model results.
The isotopic mass balance of a homogeneous lake at steady state is given by:

$$\frac{E}{I} = \frac{\delta_I - \delta_L}{\delta_E - \delta_L} \tag{1}$$





Where I is the flux of water flowing in, either as precipitation, surface runoff or groundwater, E is the evaporation flux, and $\delta_I$ and $\delta_L$ are the amount weighted isotopic compositions of of influxes and outflux, which can both be measured directly, at least for the surface inflow. $\delta_E$ is the isotopic composition of the evaporation flux leaving the lake. It is depleted in heavy isotopes relative to the lake, and depends in a large part from kinetic fractionation processes. It is calculated classically by using the

Craig and Gordon model (Craig and Gordon, 1965) :

$$\delta_E = \frac{\frac{\delta_L - \epsilon*}{\alpha} - h\delta a - \epsilon_K}{1 - h + \epsilon_K} \tag{2}$$

where $\delta_E$ is expressed as a function of equilibrium ($\alpha$, $\epsilon*$) and kinetic ($\epsilon_K$) fractionation factors, the relative humidity (h), and the isotopic composition of the atmospheric vapor ($\delta$a). The values and the calculation of these different factors were discussed extensively by several authors and are listed in the Appendix. The seasonal variations of h are obtained from regional weather

stations. The value of $\delta$a can be measured on the field either with a cryogenic trap device (Fontes et al., 1970; Krabbenhoft et al., 1990; Salamalikis et al., 2015), or more recently by laser spectrometry (Tremoy et al., 2012). Since these measurements remain relatively rare, an alternative is to assume as a first approximation that $\delta$a is in isotopic equilibrium with the precipitations, and then discuss the sensitivity of the results with respect to this assumption:

$$\delta a = \frac{\delta_P - \epsilon*}{\alpha} \tag{3}$$

$$\delta a \approx \delta_P - \epsilon* \tag{4}$$

Calculated from steady state values, or from average values integrated over the annual cycle, the E/I ratio allows us to discriminate between lakes behaving as "closed-systems", i.e. where evaporation is the only water output (E/I = 1), and "open-system" lakes from which water can flow out as surface or underground flow (E/I < 1). Cases where E/I > 1 would indicate lakes that are not in steady state, and are progressively drying out.

This approach has been used extensively in previous studies, (Zuber, 1983; Gibson et al., 1993, 2002; Mayr et al., 2007; Yi et al., 2008; Brock et al., 2009) especially for inter-comparisons between lakes hydrological budgets at a regional scale, assuming that the seasonal variations of lake level and composition could be neglected compared to the regional gradients (Gibson et al., 2002). On the other hand, in regions with high seasonal contrast in lake level between wet and dry season, it is essential to have access to annual integrated values in order to solve equation 1. Therefore, regular monitoring of the lake level and isotopic

composition, as well as of inflow and outflow measurements, are required. However, for lakes that cannot be monitored for logistical reasons, we show below that a first-order estimate of the water balance can be assessed from dry season isotopic data alone, and lake level satellite data.

Indeed, for any given lake for which the regional values of $\delta$a, h and $\delta_I$ are known, we can calculate the isotopic composition

characteristic of the closed-system situation ($\delta_{L-closed}$), by combining equation 2 with the steady state condition $\delta_E = \delta_I$ :

$$\delta_{L-closed} = (\delta_I(1 - h + \epsilon_K) + h\delta a + \epsilon_K)\alpha + \epsilon* \tag{5}$$

The isotopic composition measured during the dry season, $\delta_{L-dry}$, can then be compared to this $\delta_{L-closed}$ value. For a lake where $\delta_{L-dry} < \delta_{L-closed}$, we can deduce that the mean annual value of $\delta_L$, which is always lower than $\delta_{L-dry}$, is thus also





lower than $\delta_{\mathrm{L-closed}}$. This means that the lake is "open", with a significant outflow, and the application of equation 1 leads to a maximum value of E/I.

In a second step, a more precise E/I value can be obtained, based on seasonal lake volume variations estimated from satellite data. For this purpose, a model value of the lake's isotopic composition at the end of the wet season ($\delta_{\mathrm{L-wet}}$) is used to the mean annual $\delta_{\mathrm{L}}$. The simplest case is that of a closed-system lake, with a negligible flux of evaporation compared to the inflow during the wet season level increase. In this case:

$$\delta_{L-wet} = y\delta_I + (1-y)\delta_{L-dry} \tag{6}$$

with $y = \frac{V_{wet} - V_{dry}}{V_{wet}}$

$V_{\mathrm{wet}}$ is the maximum volume of the lake at high-stand, and $V_{\mathrm{dry}}$ the low-stand volume at the end of the dry season.

If the outflow is non negligible during the wet season, and considering constant inflow and outflow (Qin and Qout respectively) during the wet season, the equation becomes:

$$\delta_{L-wet} = \delta_I + \frac{\delta_{L-dry} - \delta_I}{(1 + \frac{\Delta V}{V_{dry}})^{\frac{1}{\gamma}}} \tag{7}$$

with $\gamma = 1 - \frac{Q_{out}}{Q_{in}}$

More complex situations with time-variable fluxes and non-negligible evaporation during infilling require using a complete finite-difference model based on a priori assumptions on the respective proportions of the different flows over time:

$$(\delta_L V)_{t+\Delta t} = (\delta_L V)_t + \delta_I Q_{in} - E\delta E - Q_{out}(\delta_L)_t \tag{8}$$

where all the terms may vary with time, and should be adjusted to fit the observed value of $\delta_{\mathrm{L}}$, knowing the time variation of the volume (V) obtained from satellite observations. Our approach using equation 5 and 6 allows us to bracket the seasonal variations of $\delta_{\mathrm{L}}$ in a much lighter way. The results of this examination are reported in table 1 for lake Iro and Fitri, with calculated values for $\delta_{\mathrm{L-closed}}$, $\delta_{\mathrm{L-dry}}$, and $\delta_{\mathrm{L-wet}}$, and chosen parameters in each case for h, $\delta_{\mathrm{a}}$ and $\delta_{\mathrm{I}}$.

### 5.1.1 Testing the method : lake Ihotry

We applied the same method to lake Ihotry (SW of Madagascar) as a test of comparison. The hydrologic cycle and isotopic data of this lake were regularly monitored and analyzed during several seasonal cycles, leading to a well constrained hydrological and isotopic model (Vallet-Coulomb et al., 2006a, b, 2008), thus constituting a valuable benchmark for our approach.

This is a small size lake (between 68 and 115 km$^2$) located in an endorheic karstified limestone watershed covering 3000 km$^2$ in South-West Madagascar (Grillot and Arthaud, 1990). The climate is semi-arid and characterized by a strong contrast between dry and rainy season. Precipitation occurs between December and March and are about 800 mm yr$^{-1}$. Climatic conditions are thus quite similar to those observed around the Chadian lakes. Lake Ihotry is fed by the Befandriana river and





associated hyporeic fluxes for 50%, and 50% by direct rainfall (Vallet-Coulomb et al., 2008). Water balance analysis shows that lake Ihotry is a closed-system lake, with evaporation representing 99% of the output (Vallet-Coulomb et al., 2006b). By contrast with Lake Chad, Iro and Fitri, this is a salted lake with a conductivity ranging between 7000 and 23 000 $\mu$S cm$^{-1}$. The calculation of $\delta_{L-closed}$ gives +0.44 ‰ and -6.1 ‰ values for $\delta^{18}$O and $\delta^2$H (Fig.6). The mean values of $\delta_{L-dry}$ measured

during the three years study are +5.44 ‰ and +26.7 ‰ for $\delta^{18}$O and $\delta^2$H, which are thus significantly higher than $\delta_{L-closed}$ (Fig.6). The value of $\delta_{L-wet}$ calculated by equation 6 (closed-system lake) are $\delta^{18}$O = -2.20 ‰ and $\delta^2$H=-19.7 ‰, very close to average isotopic composition observed in March for Lake Ihotry (Vallet-Coulomb et al., 2006b). We are thus in the situation where $\delta_{L-wet} < \delta_{L-closed} < \delta_{L-dry}$ , compatible with the diagnostic of a closed-system lake.

In the case of lake Ihotry, for which we have a regular monitoring of the water flows and isotopic composition over three years at a monthly timescale, we can push further the analysis by calculating a precise mean value of $\delta_L$ weighted by the output flows. The values we obtain for $\delta_L$ ($\delta^{18}$O = +0.58 ‰ and $\delta^2$H = -5.5 ‰) are very close to $\delta_{L-closed}$ which confirms both the closed lake conclusion, and the reliability of the method.

### 5.1.2   Iro and Fitri

The same approach was applied to lakes Iro and Fitri. The results are shown in figure 7 (green stars for Iro and orange stars for Fitri). Depending on the different assumptions that we can use for the values of the parameters $\delta_I$ and $\delta a$ (table 1), $\delta_{L-closed}$ can vary between +3 ‰ and +6 ‰ for $\delta^{18}$O, and between +11 ‰ and +30 ‰ for $\delta^2$H for lake Iro. For lake Fitri, the values range between +5‰ and +7 ‰ for $\delta^{18}$O and between +18 ‰ and +34 ‰ for $\delta^2$H. These values are in both cases definitely higher than those measured on the lakes: we are in the case where $\delta_{L-dry} < \delta_{L-closed}$ which is characteristic of open lake

systems. The resulting maximum E/I value for Iro is E/I = 0.6, calculated either with $\delta^{18}$O or $\delta^2$H. For lake Fitri, we find E/I = 0.4. However, since the sample was collected in the middle of the dry season instead of at the end, $\delta_{L-dry}$ is most probably less depleted than our measured value, and the corresponding E/I result can only be taken as a lower estimate.

This maximum E/I value can then be combined with the estimated $\Delta V/Vdry$ value from satellite data to calculate the range of $\delta_{L-wet}$ values based on equation 7 ($\delta_{L-wet}$=-0.77 ‰; -9.8‰ for $\delta^{18}$O and $\delta^2$H), keeping in mind that this is a first order

approximation, based on the simplifying assumption of constant flux and negligible evaporation during the rainy season. Finally, the arithmetic mean between $\delta_{L-dry}$ and $\delta_{L-wet}$ (0.19 ‰ for $\delta^{18}$O and -4.2 ‰ for $\delta^2$H) provides an average value of the isotopic composition of the lake, from which a ratio E/I = 0.4 is calculated for lake Iro.

These results can be confronted with the scarce hydrological information available on the studied systems. For lake Iro, based on the 1.8 m yr$^{-1}$ mean evaporation recorded at the Birao station (SIEREM), and assuming an average surface of the lake of

350 km$^2$ (halfway between the dry and wet season surfaces), we can estimate that the mean flux of vapor escaping the lake (E) is on the order of $5.10^8$ m$^3$ yr$^{-1}$. The inflow (I) is then between 17 and 31 m$^3$ s$^{-1}$ (subtracting the minor contribution of direct rainfall on the lake surface), and the outflow (Q) between 6 m$^3$ s$^{-1}$ and 11 m$^3$ s$^{-1}$. These figures are thus coherent with the average flow of 30 m$^3$ s$^{-1}$ measured on the Bahr Azoum at Am Timan station between 1953 and 1975.

For Fitri, using the evaporation of about 2 m yr$^{-1}$ , similar to that calculated on lake Chad southern pool under the same





climatic condition (Bouchez et al. 2016), we obtain an flux on the order of $16.10^8$ $m^3yr^{-1}$, and an inflow between 43 and 110 $m^3 s^{-1}$. The maximum flow rate recorded at the Ati station between 1956 and 1993 was 66 $m^3 s^{-1}$, corresponding to the maximum rainfall of 571 $mm\,yr^{-1}$ on this watershed in 1962 (DREM). By contrast, the river completely dried during the drought episodes of the 80's. However, the flows recorded at Ati are not fully representative of the total flux feeding the lake, since a multitude of hardly quantifiable streams reach the lake after the gauging station, contributing to the larger calculated flows. The calculated outflow Q is between 14 $m^3 s^{-1}$ and 36 $m^3 s^{-1}$. Since the lake is endorheic, this flux must be feeding the surrounding aquifers. However, we pointed out above that the isotopic composition of the lake water does not plot on the LEL defined by the groundwater samples (Fig.4), suggesting that the surface water is in some way disconnected from the aquifers in this case. This question is briefly mentioned again below when considering the basin-scale budget, but would require a more extended study including radioactive tracers.

### 5.1.3 Evaluation of uncertainties

In order to evaluate the sensitivity of our conclusions to the choice of values of the three principal parameters used in equation 2 (h, $\delta$I and $\delta$a), we discuss separately below the influence of each of them on the results. An illustration of this discussion is given in Figure 8, where $\delta_{L-closed}$ and E/I are plotted against $\delta$I for various values of $\delta$a and h (grey polygon).

**h**: the relative humidity is relatively constant at Ihotry (75-81% (Vallet-Coulomb et al., 2006b)), but shows a large range of seasonal variation at Iro and Fitri (25-77% and 16-75%, respectively). Average values weighted by the regional evaporation fluxes can be calculated from SIEREM data (Boyer et al., 2006), as well as their standard deviation, and the corresponding uncertainty is reported as dotted lines in Fig.8. Data from SIEREM show an annual average value of h = 52 $\pm$ 2.5 % ($1\sigma$) for Iro and h = 37 $\pm$ 3.4 % ($1\sigma$) for Fitri over 10 years, which results on very low uncertainties on E/I.

**$\delta$a**: the isotopic composition of the atmospheric moisture is the least constrained parameter because of the scarcity of measurements, leading to a poor understanding of its variations. Tremoy et al. (2012) published the only continuous time series available in the Sahelian band, carried out by laser spectrometry in Niamey (Niger). The results show a large variation during the seasonal cycle (-15 ‰ < $\delta^{18}O_V$ < -9.5 ‰) with two minima periods: one from August to September during the rainy season, and the other in January during the coldest month. The maximum occurs in May, at the end of the dry season. These variations are interpreted in terms of regional climatology, as a function of continental air masses seasonal displacement, and of the increase in convective activity during summer. In addition to these synoptic effects, atmospheric vapor may also have a significant fraction of local recycling from lake evaporation (Gat et al., 1994). For Ihotry, Vallet-Coulomb et al. (2008) showed that the lake isotopic balance implies that the isotopic composition of the atmospheric vapor varied between -13.8 ‰ and -7.8 ‰ for $\delta^{18}O$ during the annual cycle. For Ihotry and Niamey, the values of $\delta$a calculated by assuming equilibrium between precipitation and atmospheric vapor are close to the seasonal minimum value. For Ihotry, we used the average of the model $\delta$a values, weighted by the evaporative fluxes (Vallet-Coulomb et al., 2006b) (-13.14 $\pm$ 1.86 ‰ and -93.9 $\pm$ 12.9 ‰ in $\delta^{18}O$ and $\delta^2H$), and for Iro and Fitri, the average of the data from (Tremoy, 2012), again weighted by the regional evaporative



fluxes (-12.99 $\pm$ 1.7 ‰ and -93.9 $\pm$ 14.0 ‰). The use of Niamey's data for these two lakes was previously used on the Lake Chad by Bouchez et al. (2016), and is justified by the strong correlation between the $\delta^{18}$O measurements in Niamey and the LMD-ISO model outputs for the sahel band (Tremoy et al., 2012). The resulting range of uncertainty is illustrated in Fig.8 in the case of Iro, and listed in Table 1 for Ihotry and Fitri.

$\delta$I: The assumption of $\delta$I $\approx$ $\delta$P is much more questionable for large catchments where the isotopic composition of the rivers may be more or less marked by evaporation. In the case of lake Iro and Fitri, in absence of isotopic data for their main river inflows of lake Iro and Fitri, we chose the most depleted groundwater value, close to the intersection between the GMWL and the LEL, as our first-guess estimate for $\delta$I. This constitutes a minimum value and thus gives a maximum result for E/I

(Table 1). However, recent studies of the Chari-Logone have shown that the weighted annual average composition of the river is distinctly enriched compared with the intercept between the LEL and the LMWL (c.a 2 ‰) (Mahamat Nour et al., 2017). Assuming a similar shift for the Bahr Salamat and Batha, and estimating again the uncertainty from the averages weighted by the river fluxes, we obtain lower E/I values of 0.65 $\pm$ 0.3 for Iro and 0.41 $\pm$ 0.2 for Fitri (green circle in Fig.8, Table 1), which thus constitute our preferred estimate, until further characterization of these rivers.

## 5.2 Water balance at the catchment scale

In parallel of the characterization of the lakes hydrological regime, it is also important to evaluate possible hydro-climatic inferences on their catchment, especially for the influence of evaporation and transpiration on the water balance.

At this scale, we underlined the existence of the three different LEL, close but distinct from each other for the three sub-catchments (Tchad, Iro and Fitri), which thus constitute independent entities. The parallel slopes in $\delta^2$H versus $\delta^{18}$O can be

readily explained by a similar set of values for the different parameters of the Craig and Gordon model at the catchment scale. On the other hand, interpreting the different intercepts for the three evaporation lines is less straightforward.

The first assumption that comes to mind is to assign a specific rainfall isotopic composition to each catchment, corresponding to the intercept with the GMWL. This implies that small climatic mean differences exist between the catchments, that remain stable at the catchment scale on time scales comparable to the water residence time. Such differences could be attributed to

different orographic effects among the catchments, or to a North-South gradient (Terzer et al., 2013). This interpretation can be seen as coherent with the difference noted between the weighted average rainfall isotopic composition at N'Djamena between 1962 and 2015 (-3.5 ‰ for $\delta^{18}$O), and Iro and Fitri's evaporation lines.

However, an alternative hypothesis could be to ascribe the difference to a different partition between transpiration and evaporation in the different catchments. An attribution method based on solving the three end-members equation between transpiration

(T), evaporation (E), and outflow (Q), using the property that T does not fractionate the isotopes of water, while E induces a strong fractionation (Gibson et al., 2008; Jasechko et al., 2013).

The solution of the equation for a steady state catchment without additional inflow from upstream, is :

$$T = \frac{P(\delta_P - \delta_E) - Q(\delta_L - \delta_E)}{\delta_T - \delta_E} \qquad (9)$$



P is the average precipitation on the catchment, $\delta_E$ is the isotopic composition of the evaporation flux, obtained again with the Craig and Gordon model from the lake isotopic composition. ($\delta_L$) is considered as representative of all the surface water over the catchment. For $\delta_T$, Jasechko et al. (2013) calculate the average rainfall composition over the seasonal cycle, weighted by the NDVI index (Normalized Difference Vegetation Index) (Curran and Steven, 1983; Defries and Townshend, 1994) used

as a transpiration intensity proxy. In all cases where the rainfall composition varies during the season, and if the vegetation growth cycle is out of phase with precipitation, the calculation does lead to different values between $\delta_T$ and $\delta_P$. This is indeed the case for the N'Djamena IAEA station data set, as illustrated in Fig.9. We observe a large variation from -10 ‰ to +10 ‰ in $\delta^{18}O$ between the beginning and the end of the rainy season, the heavy rainfall between July and August being the most depleted due to the amount effect (Dansgaard 1964). The NDVI is lagging rainfall by about a month, and $\delta_T$ and $\delta_P$ (assuming

interception value between Iro MWL and the GMWL is $\delta_P$ ) are thus significantly different (-1 ‰ versus ≈-3 ‰ in $\delta^{18}O$). Moreover, the seasonal cycle of $\delta_P$ may explain the apparent disconnection between the lake and the aquifer at Fitri, if the aquifer is preferentially recharged by these heavier summer rains, thus imprinted by their more depleted signature.

The mass balance consequence of the difference between $\delta_T$ and $\delta_P$ is then that the composition of the water entering the lake-aquifer system ($\delta_I$) is different from $\delta_P$, as observed for Iro and Fitri. In principle, the T/P ratio could be calculated from

this difference (0.37 for Iro and 0.43 for Fitri). However this result is clearly strongly model-dependent, based on the type of assumption adopted to estimate $\delta_T$, directly related to the assessment of water depth removal by the vegetation and thus residence time within the soil before respiration (Dawson, 1996).

Alternatively, the approach of Jasechko et al. (2013) was to solve equation 9 using values measured at the catchment outlet for Q, and at local weather stations for P. The results are presented in table 2 for the three lakes studied here, and illustrated in

Fig.10. For the catchment of lake Iro, we took the measured flow rate of the Bahr Salamat river as representative value of Q, and a flow rate equal to zero for the endorheic catchments of Fitri and Ihotry. The results suggest that transpiration (T) represents half of the water balance of the catchment in every scenario (0.42 < T/P < 0.56). However, a closer examination of the weight of the different terms in equation 9 stresses the limitations of this method already highlighted by several authors (Schlaepfer et al., 2014; Jasechko et al., 2013; Coenders-Gerrits et al., 2013). First, using the lake isotopic composition as a representative

value for the entire surface water of the whole catchment is highly speculative: indeed, the balance between evaporation and runoff is very unlikely to be the same between the lake itself and the soil and streams water of the watershed. Moreover, the water fluxes transiting through the lakes represent only a small fraction of the rainfall falling on the entire catchment (<1 %, 10 % and 3 %, respectively, for Iro, Fitri and Ihotry). We can illustrate the strong sensitivity of the calculated T/P ratio upon the $\delta_L$ value: for example, if the mean isotopic composition of aquifer is taken as representative value in the case of Iro, instead of

that of the lake, the T/P ratio increases from 0.42 to 0.77. By comparison, the $\delta_T$ value is a much less stringent parameter in this case.

This representativeness problem is even more clearly illustrated by the Fitri catchment. We have shown above that the value of $\delta_L$ , and thus of $\delta_E$, correspond to E/I = 0.4. It is thus incoherent to associate this value of $\delta_L$ with a Q value of zero, typical of an endorheic basin, and this shows that the results obtained on the lake cannot be extrapolated to the catchment scale.





Finally, another serious problem appears for the catchment of a closed-system lake, as Ihotry, for which equation 9 becomes:

$$T = \frac{P(\delta_P - \delta_E)}{\delta_T - \delta_E} \tag{10}$$

In this case, as discussed above, the water balance implies that $\delta_I = \delta_E$ . We are thus faced again with a simple mixing equation between P, T and I, the solution of which is totally subordinated to the choice of $\delta_T$, and even becomes totally undetermined in the case of $\delta_T = \delta_P$ .

## 6 Conclusions

Our study illustrates the possibility of obtaining quantitative constrains on isolated inter-tropical lakes, in absence of hydrological monitoring, by coupling dry season isotopic data with satellite imagery. We validated the method on lake Ihotry in Madagascar, by confirming the closed-system behavior of this endorheic lake.

For lake Iro and Fitri, two sub-catchments in the lake chad basin, we obtained E/I ratios of 0.6±0.3 and 0.4±0.2, and evaluated a maximum range of uncertainty depending on the choice of parameters values.

For both cases, the model inflow rate is on the same order as the flow measured upstream, on the Bahr Salamat for Iro and on the Batha for Fitri. For the latter, there is no identified surface outflow, the outflow is thus necessarily feeding the groundwater reservoir. In both cases these water fluxes represent a small part of the total precipitation on the basin (<1 % and 10 %).

These model E/I ratios and water fluxes are coherent with the freshwater character of these lakes, as for the southern pool of the lake Chad itself, in spite of the high evaporation rate of the Sahel region.

In terms of isotopic composition, the three sub-catchments show distinct evaporation lines with similar slopes but slightly different intercepts. The mean rainfall composition at N'Djamena for 1962-2015 plots on the line of lake Chad data. The data from the lake and from the surrounding groundwater plot on the same line for Iro, while they are slightly offset for Fitri, suggesting some disconnection between surface and aquifer waters in this second case. These differences may be attributed to small climatic variations between the catchments. An alternative explanation can be the contribution of the rain to groundwater recharge and river runoff along seasonal variations. This can be the consequence of the isotopic partitioning between evaporation and transpiration fluxes at the catchment scale. Both assumptions would deserve regular monitoring of the lakes and regional precipitation.

Our results on the two Chadian lakes and on lake Ihotry enable us to test the method proposed by Jasechko et al. (2013) to estimate vegetation transpiration fluxes at the catchment scale. Although broadly compatible with the conclusion that transpiration dominates the water balance of these catchments, the results can only be considered as semi-quantitative, owing to the large uncertainties resulting from the questionable representativeness of the lakes for the total surface water of the catchment, especially for endorheic and closed basins like Fitri and Ihotry. In particular, this method does not permit a detailed comparison between sub catchments under different climates.

As a whole, this study confirms the great interest of these lakes as miniature analogs of the lake Chad itself, as shallow fresh-




water lakes with an outflux toward the aquifer for Fitri, making them important targets in future set up of any large-scale monitoring program of the hydro-climatic evolution in the Sahel region.

*Code availability.*  TEXT

*Data availability.*  TEXT

*Code and data availability.*  TEXT

## Appendix A:  Parameters of Craig and Gordon model

The evaporation isotopic composition is classically calculated by using Craig and Gordon model (Craig and Gordon, 1965) :

$$\delta_E = \frac{\frac{\delta_L - \epsilon*}{\alpha} - h\delta a - \epsilon_K}{1 - h + \epsilon_K} \tag{A1}$$

$\delta_L$ is the lake isotopic composition, $\delta a$ is the atmospheric isotopic composition and h the relative humidity in %. $\alpha$ is the
equilibrium fractionation factor between liquid water and water vapor, and depend on temperature. Some authors proposed closed formulas depending on temperature (Majoube, 1971; Horita and Wesolowski, 1994). $\epsilon*$ is the total fractionation coefficient : $\epsilon* = \alpha - 1$.

$\epsilon_K$ is the kinetic fractionation : $\epsilon_K = C_K(1 - h)$. $C_K$ can be calculated as a function of turbulent parameters : $\epsilon_K = \theta.n.C_D$. With n=0.5 for an average turbulent flow(Gonfiantini, 1986). $C_D$ has been experimentally determined by Merlivat (1978) :
$C_D(^{18}O)$=28.5 ‰ and $C_D(^2H)$=25.1 ‰. $\theta$ can be equal to one for a small water surface, where evaporation fluxes don't disrupt air humidity. It has been demonstrated that $\theta = 0.88$ for American northern lakes and $\theta = 0.5$ for mediterranean see (Gat et al., 1994; Gat, 1995, 1996). For this study we chose $\theta = 0.5$ as used for lake Chad and discussed in Bouchez et al. (2016).

*Author contributions.*  TEXT

*Competing interests.*  TEXT

*Disclaimer.*  TEXT





*Acknowledgements.* This work was part of FSP GELT (Fonds de Solidarité Prioritaire "Grands Ecosystèmes Lacustres Tchadiens") program funded by the French Ministry of Foreign Affairs. We thank people from the Iro and Fitri regions as well as the local authorities of Chad for support and collaboration during fieldword. Special thanks to Hassan Mahamat Absakine, sultan of Fitri and the chief of Iro to offer their hospitality to the GELT teams. P.Deschamps and F.Sylvestre specifically acknowledge O.D Hont and F.Gianviti from the SCAC of the French Ambassy in Chad as well as the CNRD to have made possible fieldtrips in Iro and Fitri areas. We also thank C.Raimond, D.Zakinet, L.Gonzales and K.Nkouka for their support in the field organization. This work is a contribution to Labex OT-Med (n°ANR-11-LABX-0061) and has received funding from Excellence Initiative of Aix-Marseille University - A*MIDEX, a French "Investissements d Avenir" programme.





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

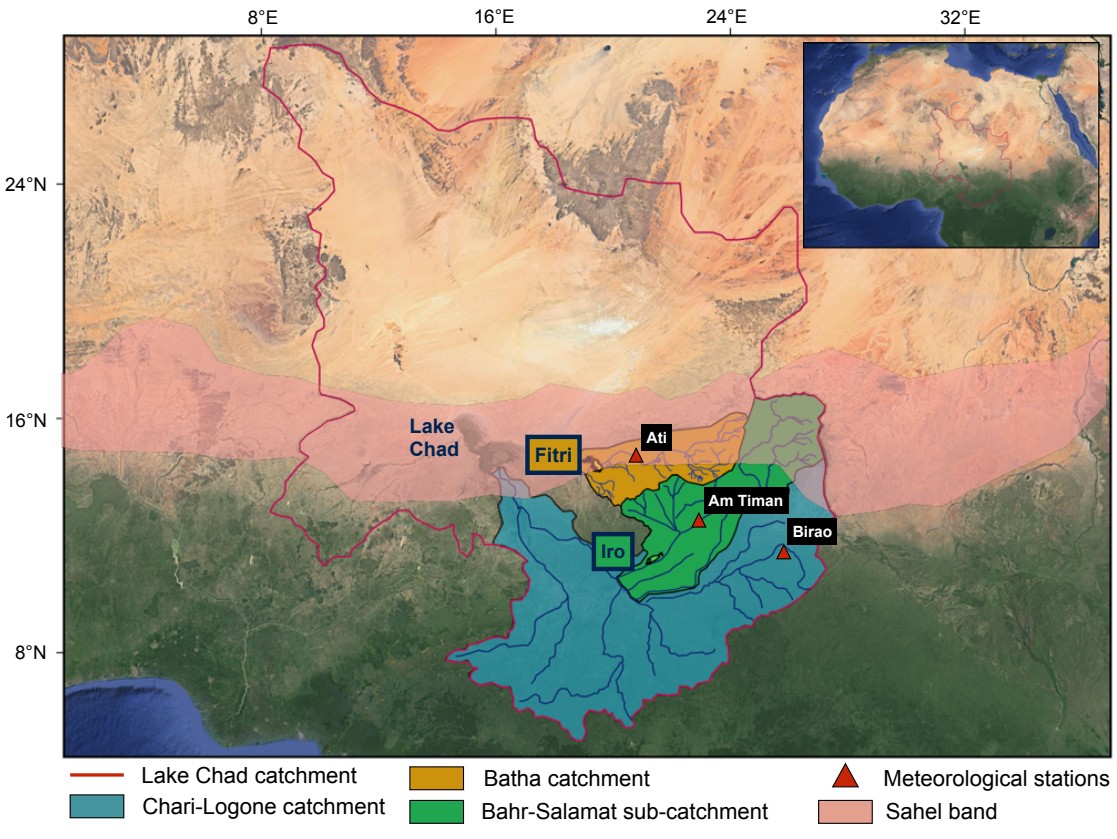

**Figure 1.** Lake Chad Basin (red line) is crossed by the Sahel band (pink) which delineates the arid Northern part and the humid Southern part. It is constituted by two main catchments : the Chari-logone feeding the lake Chad (blue) and the Batha feeding the lake Fitri (orange). The Bahr Salamat (green) is a sub-catchment of the Chari-logone and feeds the lake Iro. Data from three meteorological stations (red triangles) have been used for this study.



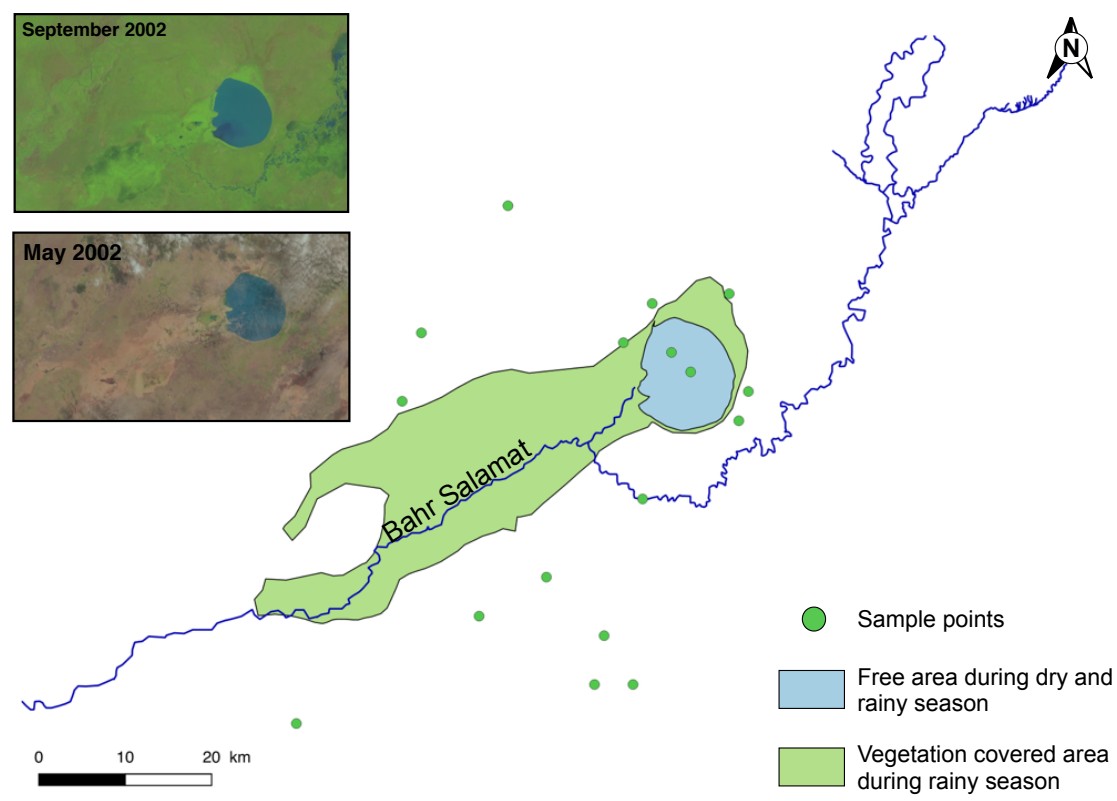

**Figure 2.** Landsat 7 pictures of lake Iro during the wet season (September) and at the end of the dry season (May). The states of the lake are represented in blue for free water and in green for vegetation covering the flood plain during the rainy season. Only a part of the Bahr Salamat feeds the lake. Green points represent the sampled water during April 2015.

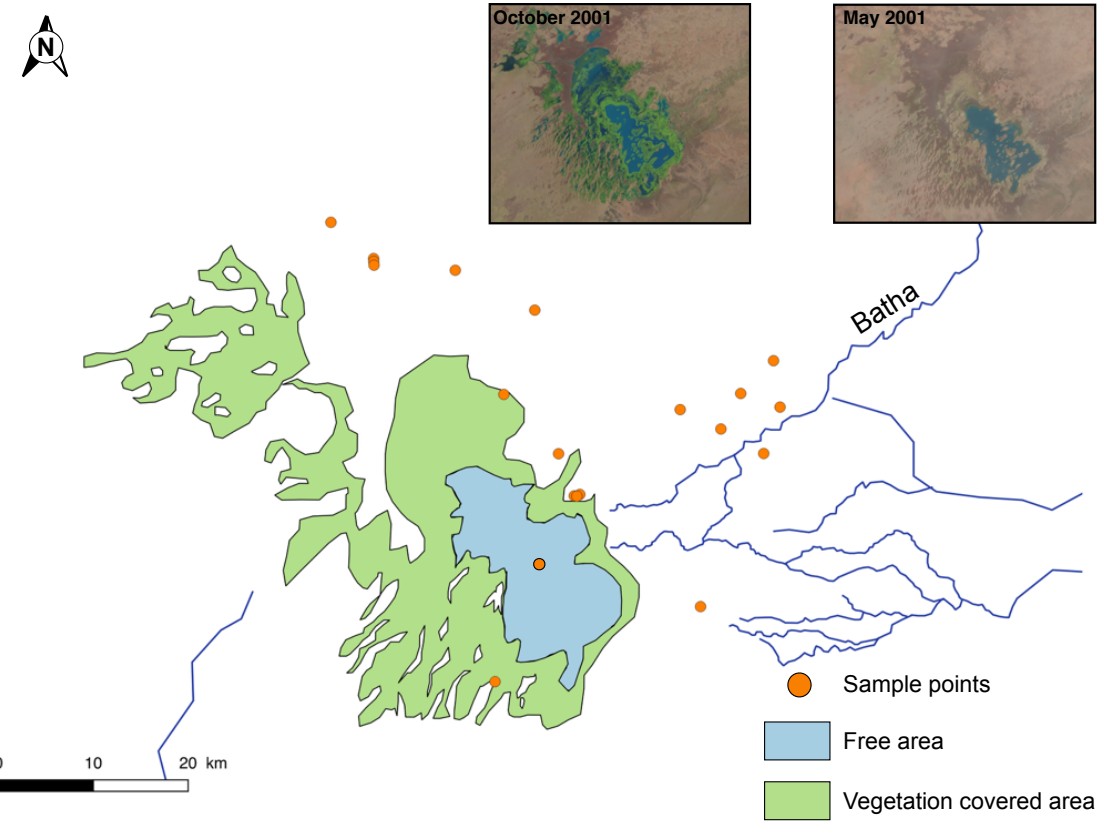

**Figure 3.** Landsat 7 pictures of lake Fitri during the wet season (October) and at the end of the dry season (May). The states of the lake are represented in blue for free water during wet and dry season, and in green for vegetation covering a largest area during the rainy season. The Batha is the main tributary, and orange points represents the sampled water during January 2015.





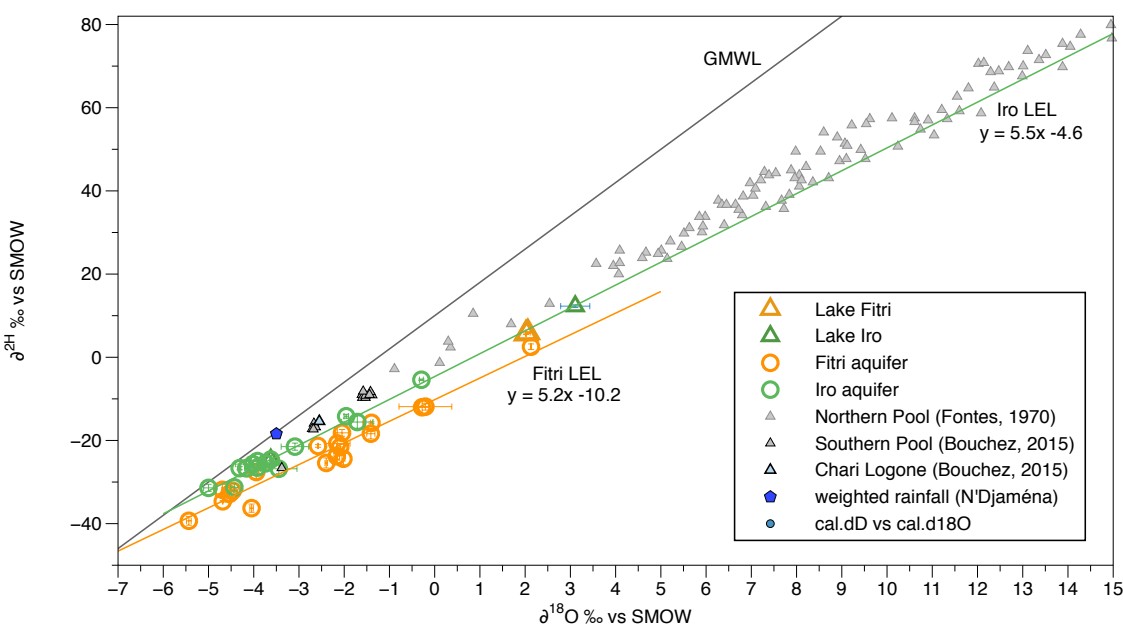

**Figure 4.** Iro's groundwater (green) form a Local Evaporation Line (LEL) on which are ploted the data from Lake Iro and Fitri (triangles). Fitri's groundwater forms another LEL below that of Iro. In grey, the lake Chad data forming its own LEL (y=5.2x+1) is above the two previous lines. The weighted rainfall (from N'djamena IAEA station) is in blue and the associated Local Meteoric Water Line (LMWL) equation is y=6.3x+5.





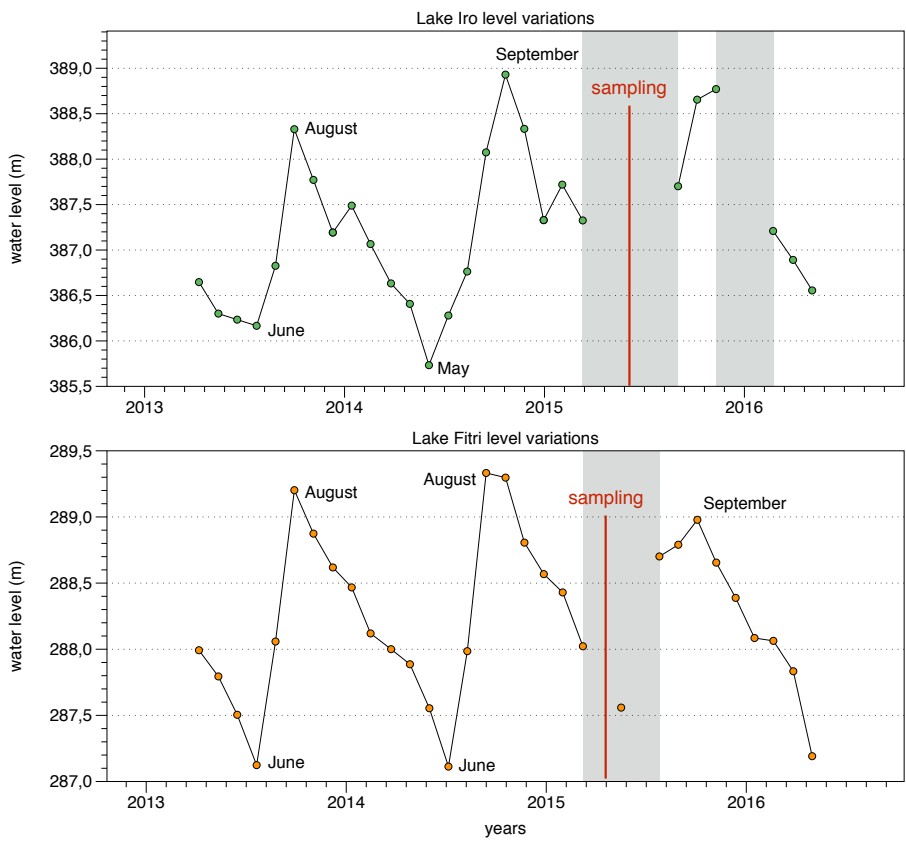

**Figure 5.** Lake level variations from altimetry data (SARAL satellite) for lake Iro (in green) and lake Fitri (in orange). Data cover the period from 2013 to the beginning of 2016, the grey zones represent a lack of data, and the red line the period of sampling. Lake Iro level variations show a maximum of 3.5m during 2014, and a maximum of 2.5 m for lake Fitri. The maximum level is observed around august and september, and the minimum in may - june for both lakes.



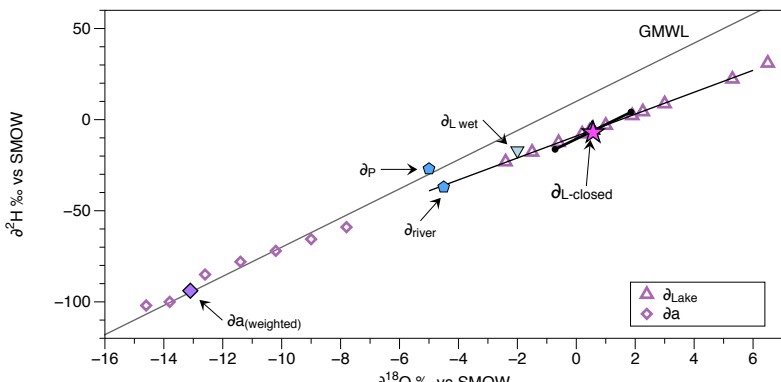

**Figure 6.** Purple triangles are the lake Ihotry isotopic composition over a year, and the purple diamond are the atmospheric isotopic composition ($\delta$a) over a year (the colored one represent the $\delta$a weighted average). $\delta_P$ is the rainfall isotopic composition above lake Ihotry and $\delta_{river}$ is the Befandriana river isotopic composition which is feeding the lake Ihotry. The pink star represents the $\delta_{L-closed}$ calculated according equation 4. The blue triangle is the $\delta_{L-wet}$ calculated by equation 5. The black line around $\delta_{L-closed}$ represents the associated uncertainties.





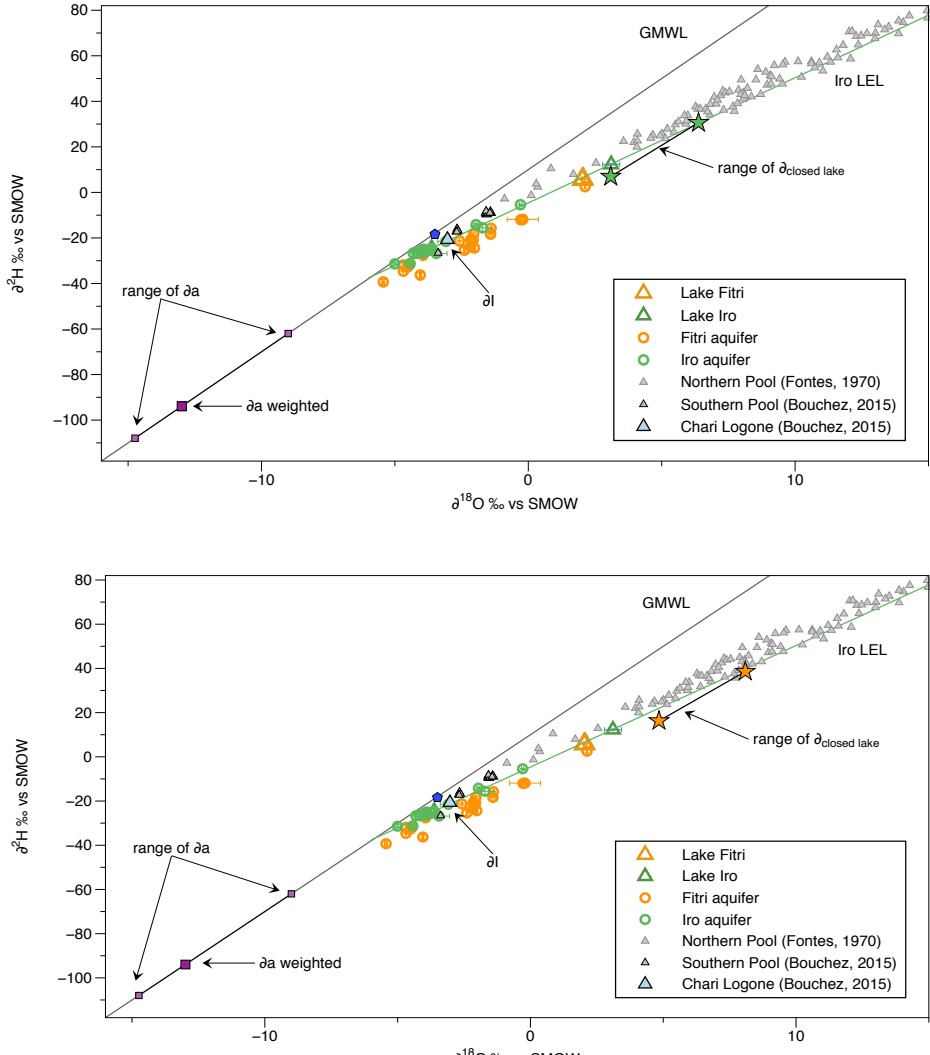

**Figure 7.** Representation of $\delta_{L-closed}$ variations for lake Iro and lake Fitri (green and orange stars respectively). They are slightly below Iro's Local Evaporation Line (LEL) in green formed by Iro's groundwater (green circles). Lake Iro and Fitri (green and orange triangles) plot on this LEL, but Fitri's groundwater (orange circle) are just below. Purple points represent the atmospheric isotopic composition ($\delta$a). $\delta_I$ is the isotopic composition entering the system for Iro surface water and groundwater and for Fitri surface water only. We took $\pm 1\ \sigma$ on $\delta$a and $\delta_I$ to represents the range of variations of $\delta_{L-closed}$.

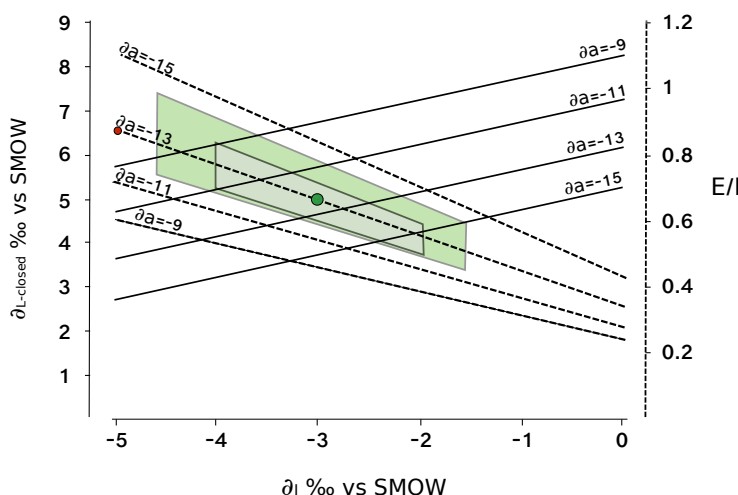

**Figure 8.** Final $\delta^{18}O$ uncertainties on $\delta_{L-closed}$ and E/I by taking $\pm 1\,\sigma$ on $\delta a$ and $\delta_I$. The green point represent the E/I and $\delta_{L-closed}$ average value, and the green polygon is the associated uncertainty. $\delta a$ is the mean annual value (Jasechko et al., 2013) weighted by evaporation flux (DREM data) and $\delta_I$ is the Chari-logone mean annual value weighted by flow rate (Bouchez et al., 2016). The red point represents the $\delta_{L-closed}$ and E/I values for $\delta_I$ taken at the intersection between Iro's LEL and the GMWL. The grey polygon represents the E/I uncertainties by taking h$\pm5$%.



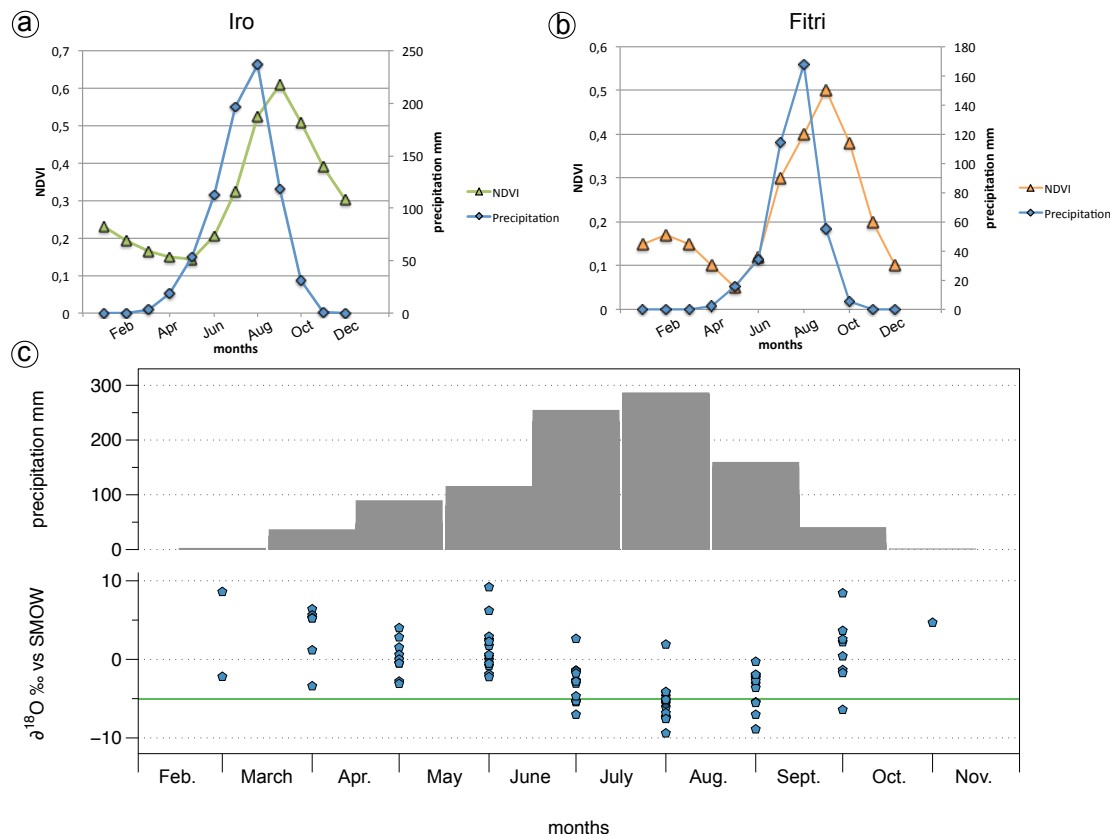

**Figure 9.** a) and b) represent the monthly average precipitation (blue) over a year at Am Timan and Ati station for Iro and Fitri respectively. Green and orange data represent the NDVI (Normalized Difference Vegetation Index) evolution over a year. For both catchments, a one month delay between the maximum precipitation and the maximum NDVI is observed. c) The grey bars represent the monthly average precipitation. The blue points represent the precipitation isotopic signature evolution ($\delta^{18}O$) month by month and the green line is $\delta_I$ (taken at the intersection between GMWL and LEL). The maximum of precipitation correspond to a depleted isotopic signature (Dansgaard effect).





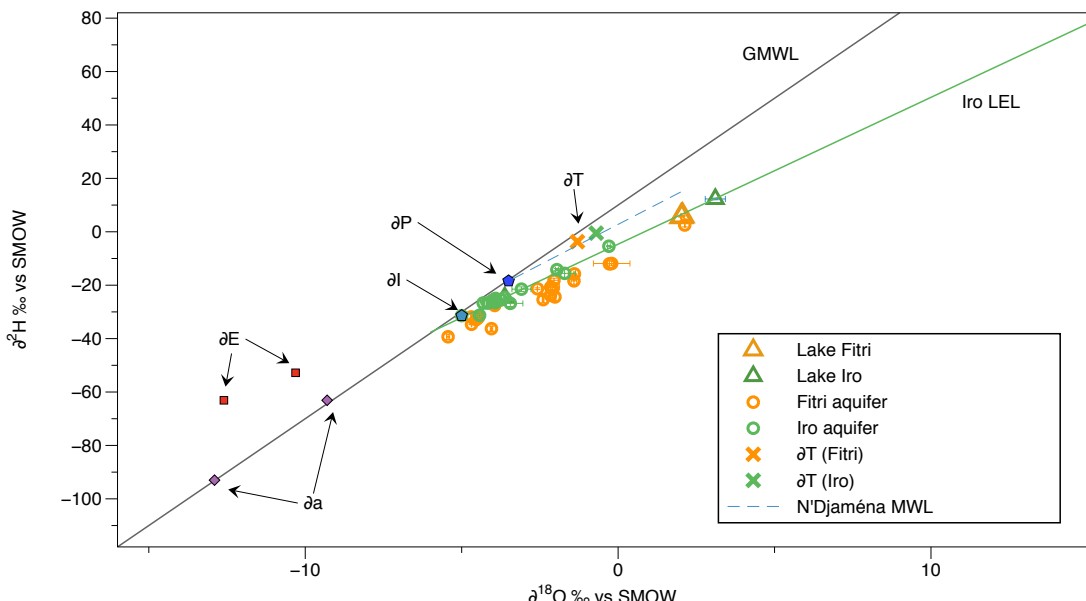

**Figure 10.** the green and orange circles represent respectively Iro and Fitri groundwater and triangles are lake water. $\delta_I$ is the water entering the system for Iro surface and groundwater and for Fitri surface water only. $\delta_P$ is the weighted annual rainfall at N'djamena and $\delta_T$ is the transpiration isotopic composition calculated according Jasechko et al. (2013). $\delta_a$ represents the minimum and the maximum value of atmospheric isotopic composition and $\delta_E$ is the minimum and the maximum value of evaporated isotopic composition calculated according the Craig & Gordon model.





**Table 1.** Uncertainties on atmospheric composition ($\delta$a), isotopic inflow ($\delta_I$), $\delta_{L-closed}$ and E/I, for the three study lakes. We took $\pm\,1\,\sigma$ on $\delta$a and $\delta_I$.

| | | h% | $\delta$a ‰ | $\delta_I$ ‰ | $\delta_{L-closed}$ ‰ | E/I |
|---|---|---|---|---|---|---|
| **IRO** | $\delta^{18}$O | 0.5 | -12.99 $\pm$1.7 | -3 $\pm$1.48 | 4.74$\pm$1.65 | 0.65$\pm$0.3 |
| | $\delta^2$H | | -93.9$\pm$14.7 | -20$\pm$7.8 | 18.7$\pm$11.8 | 0.73$\pm$0.4 |
| | $\delta^{18}$O | 0.5 | -12.99 $\pm$1.7 | -5 | 3.72$\pm$0,88 | 0.86$\pm$0.1 |
| | $\delta^2$H | | -93.9$\pm$14.7 | -31 | 12.8$\pm$7.3 | 0,97$\pm$0.3 |
| **FITRI** | $\delta^{18}$O | 0.4 | -12.99 $\pm$1.7 | -3 $\pm$1.48 | 6.46$\pm$1.62 | 0.41$\pm$0.2 |
| | $\delta^2$H | | -93.9$\pm$14.7 | -20$\pm$7.8 | 27.4$\pm$11.2 | 0.41$\pm$0.2 |
| | $\delta^{18}$O | 0.4 | -12.99 $\pm$1.7 | -5 | 5.24$\pm$0,70 | 0.57$\pm$0.1 |
| | $\delta^2$H | | -93.9$\pm$14.7 | -31 | 20.2$\pm$6.0 | 0.62$\pm$0.1 |
| **IHOTRY** | $\delta^{18}$O | 0.78 | -13.14 $\pm$1.86 | -5.45$\pm$1.87 | 0.44$\pm$1.88 | 1.12$\pm$0.7 |
| | $\delta^2$H | | -93.9$\pm$12.9 | -27.5$\pm$12.7 | -6.1$\pm$13.9 | 1.12$\pm$0.7 |





**Table 2.** Values used for transpiration calculation according to equation 9 for the three lakes study. Other data from Lake Chad come from (Bouchez et al., 2016), and data from Ihotry are from (Vallet-Coulomb et al., 2008).

| | | $\delta_L$ | $\delta_P$ | $\delta_T$ | $\delta a$ | $\delta_E$ | T°C | ha% | S km$^2$ | I mm yr$^{-1}$ | T mm yr$^{-1}$ | T % |
|---|---|---|---|---|---|---|---|---|---|---|---|---|
| Lake Chad | $\delta^{18}$O | 8.2 ± 3.6 | -3.2 ± 1 | -1.8 ± 1.9 | -9.3 ± 3 | -3.4 | 27 | 36 | 976 300 | 739 | 111 | 14 |
| (Jasechko, | $\delta^2$H | 45 ± 19 | -17 ± 9 | -8 ± 14 | -69 ± 39 | -13.5 | - | - | - | - | -464 | -61 |
| 2013) | | | | | | | | | | | | |
| Lake Chad | $\delta^{18}$O | -2.6 | -3.5 | -1.8 ± 1.9 | -12.99 | -18.5 | 27 | 40 | 760 000 | 739 | 981 | 89 |
| (other data*) | $\delta^2$H | -16 | -18.4 | -8 ± 14 | -93.9 | -85.3 | - | - | - | - | 891 | 81 |
| Iro | $\delta^{18}$O | +3.11 | -5 | -0.7 | -9 | -10 | 27 | 50 | 195 000 | 765 | 390 | 52 |
| | $\delta^2$H | +12.3 | -31 | -0.5 | -69 | -57 | - | - | - | - | 332 | 44 |
| Fitri | $\delta^{18}$O | +2.04 | - | -1.3 | -13 | -10 | - | 40 | 96 000 | 360 | 203 | 56 |
| | $\delta^2$H | +5.8 | - | -1.3 | -94 | -53 | - | - | - | - | 153 | 42 |
| Ihotry | $\delta^{18}$O | 0.58 | -5.45 | -3.8 | -13.14 | -5.18 | 27 | 80 | 3000 | 842 | 68 | 8 |
| | $\delta^2$H | -5.5 | -27.5 | -20 | -93.9 | -25.6 | - | - | - | - | 357 | 42 |