# Peer review of "Unraveling the hydrological budget of isolated and seasonally contrasted sub-tropical lakes"

_Hydrology and Earth System Sciences, 2018_

## Referee Comment (RC1) · M. Coenders-Gerrits (Referee) · 15 Jun 2018

The authors present a study where they use stable water isotopes to estimate the evaporation from Lake Chad. Although the applied method is appropriate, I miss the point why stable isotopes are used. For determining in and outflows of a lake I would think simple discharge and water level measurements would work too. And the latter, uses much less assumptions then the isotope method. Why do the authors use stable water isotopes? I assume there is a valid reason, however it's not clearly stated in the manuscript. Maybe it's related to my second concern on the paper: what is the objective? The lack of a clear objective, causes that it seems the paper is 'all over the

place' and is difficult to follow and understand. It's about determining in- and outflow. It's about testing an isotope method, and about an attempt to say something about the effect of vegetation. Therefore, I recommend to rewrite the manuscript in such way that it follows the structure: research question => objective => method =>result => discussion => conclusion; and try to put more focus in the manuscript. Lastly, the English language is sometimes also not correctly used. I recommend to ask a native speaker to check the manuscript.

Specific comments
-P1 L15: abbreviations E/I are not explained
-P5 L7: these values are average values?
-P5 L12: in semi-arid zoneS
-P5 L22: remove space after 2015
-P5 L24: add permille symbols
-P5 L30,31,32: remove space before : and .
-P6 L10: "we think that..". Things that you think, should not be in the results section, but belong to the discussion section
- Section 5.1: should be in the Methodology section.
-P6 L30: why can assume steady-state conditions of the lake? Please elaborate.
-P7 L11-13: any reference for this statement? To me this seems to be a bold statement since $d_a$ are relatively light isotopes, while precipitation is in the beginning more heavy.
-P7 L12: precipitations => precipitation
-P7 L30: what is meant by a closed-system?
-P12 L24: I think you refer to the wrong paper. This should be the correct one.
Coenders-Gerrits, A.M.J., Van der Ent, R.J., Bogaard, T.A., Wang-Erlandsson, L., Hrachowitz, M., and Savenije, H.H.G. (2014): Uncertainties in transpiration estimates, Nature, 506, E1-E2.
-P29 fig 9c: unit of precipitation is mm/month.

---

## Author Comment (AC1) · 25 Jun 2018

We thank the reviewer for her comments on our manuscript. However, we strongly believe that contrary to her opinion, our objectives are clearly stated in the abstract and in the introduction, i.e. to derive quantitative constraints and uncertainties on the hydrological budget of tropical lakes and watersheds from a limited isotope data set, when classical hydrological investigation by comprehensive flux monitoring cannot be deployed for compelling logistical reasons as is the case in remote desert or sub-tropical regions (with considerable security risks) in central Africa.

We are also most surprised that the rationale for using isotopes in such a context is

questioned, as it is widely considered a classical approach documented by numerous classical as well as recent studies.

In addition to providing insights into the hydrological functioning of a largely unknown but important hydrological setting, a further novelty is the rigorous assessment of uncertainties on the isotopic budgets which is still largely overlooked in the literature - especially in hydrological systems subject to large seasonal variations - and this remains crucial for water management issues in particular in semi-arid regions like the Sahel. For instance, just as we submitted our paper, Cui et al. (2018, Hydrological Processes 32 (3): 379–87) documented the wide variety of impacts of the seasonal variation on various lakes isotopic budgets, based on a review of previous studies where these lakes could be extensively monitored. Our approach is entirely complementary, by inferring the annual range of isotopic variations from dry-season measurements alone, completed by satellite altimetry.

We regret that our conclusion was not appreciated, in which we suggest that studying lakes of relatively small-scale, such as Iro and Fitri, in huge and complex basins as that of lake Chad, is both feasible and useful as a complement to studying the main lake itself.

Finally, we thank the reviewer for her general remarks which will help to improve the clarity of our paper, although noting that they remain of relatively second order with respect to her general poor opinion on our work.
* * *

---

## Referee Comment (RC2) · T. Steenhuis (Referee) · 23 Jul 2018

The manuscript is about a water balance in two lakes in the Lake Chad basin. It claims that it has wider applications to the water balance of Lake Chad

The English of the manuscript is pretty good. The form of the manuscript is unconventional and difficulty to follow: After reading the manuscript for the first time, I was not sure what the authors wanted to convey. Reading it twice more, I think that the authors using stable isotope analyses on two lakes that they visited each one or two times for a relatively short period during the dry monsoon phase (it is not accessible during the rain phase). I also understand that the authors tested the method for a lake

elsewhere on Madagascar. The study was mentioned in the abstract and then in the results section. I could not find the name of the site and its description in the material and methods section. I am not sure what the results are. It seems that the proposed method has large uncertainties. This should not be a surprise when trying from point measurements to derive the spatial distribution of aquifers and sources.

This manuscript might contain excellent research. I really cannot judge this from the paper as written. I read somewhere when preparing for class on scientific writing, for an author to get recognition good research that it needs to be written up so that it can be recognized.

The problem with the paper is that it has no clear objective as noted by the first reviewer. The response of the authors to this remark is

"However, we strongly believe that contrary to her opinion, our objectives are clearly stated in the abstract and in the introduction, i.e. to derive quantitative constraints and uncertainties on the hydrological budget of tropical lakes and watersheds from a limited isotope data set, when classical hydrological investigation by comprehensive flux monitoring cannot be deployed for compelling logistical reasons as is the case in remote desert or sub-tropical regions (with considerable security risks) in central Africa.as such. "

I am sorry, but I cannot find this objective in the introduction. The words "goal" or "objective" does not appear anywhere. The closest is the following statement

"Studies of inter-tropical lakes raise specific difficulties, mainly related to intrinsic characteristics such as the extremely high evaporation rates, added to high transpiration of aquatic vegetation, huge seasonal variations of fluxes and lake level, and seasonal changes in hydrologic configuration resulting from ephemeral swamps and humid zones flooded during the wet season. Moreover, an even more compelling problem stems from the logistical impossibility to reach the field sites during the wet season, when the tracks network is virtually impracticable during several months. As

a consequence, we are generally missing critical data for the recharge period, to infer seasonal ranges of variation, and thus annual average values."

Moreover, the introduction should be written in such a way that the objectives follows logically (based on references in the literature that such a study is needed. The citation above is clearly an attempt to do this. However, it falls short. There are no references. Moreover, the evaporation rates over lakes are equal to the potential rate and easy to calculate. Transpiration of aquatic plants have been rumored to increase transpiration, but I have still to see a reference to it. In addition, there are plenty of studies who have attempted and successfully done on lakes in the tropics. The study of Liebe cited in the manuscript is one and I am aware of other studies we have done in Ethiopia and Haiti and the Dominican Republic that looked at lake level changes. Without doubt there are many more. So, there are ways to get around "the tracks network is virtually impracticable" (must mean: the roads are inaccessible). (just use science citation index to find these and other references. I signed the review)

Despite all "the big talk" in the manuscript about Lake Chad, climate change and future needs, this study is about two small lakes in the Lake Chad basin and one in Madagascar where stable isotope analysis is used to do a water balance. Stating this as an objective would helped greatly. It would also be helpful to introduce most of the studies mentioned in the discussion in the introduction. Especially a section on stable isotope analysis and how is applied in this study would have helped greatly. In addition, the reader would be interested to know why this study is better than all the other studies carried out on stable isotope analysis and other water balance studies.

As I indicated above, for me to understand this article well, I would have like to have seen a section either in the introduction or part of the material and methods on the theoretical background of the stable isotope method and its uncertainties. Currently the uncertainties are piecewise introduced in the discussion, making it nearly impossible to grasp the concepts (at least for me). This is not to say that an experienced person in the field of stable isotope analysis would not understand it, but there is wider audience

reading HESS.

Good practice is that the Material and Methods discusses all locations and methods preferable in the same order as they appear in the results. I noted already before that the study in Madagascar comes as a complete surprise in the results section and without context for this site. I really do not understand it within a reasonable time frame to do this review. I do not have the patience nor the time to look up the previous studies. Moreover, the result section has other parts that are material and methods for example

"The local meteoric water line (LMWL) closest to our study sites is given by the rain samples collected at the IAEA station of. . . . . . . . . . ."

An exact comparison between the lake level measurement and the stable analysis is attempted in the last section of the paper. It is nearly impossible for me to understand it (I am not an expert in the field of stable isotope analysis).

In summary, I agree with reviewer 1, that a complete rewrite is needed. I do not agree with the response of the authors as noted above. Please follow standard writing guidelines. The reader is used to those and makes the manuscript readable. After this is done the manuscript can be evaluated on its scientific qualities

Tammo Steenhuis

---

## Author Comment (AC2) · 21 Aug 2018

We thank the reviewer for his work on our manuscript, and for his detailed prescriptions. The reviewer criticized the lack of clear presentation of our objectives, and the structure of the manuscript that he considers difficult to follow. He also did not understand in what purpose we used stable isotopes, and to what extent our study was different from previous ones on remote areas. He has specific recommendations. First, he suggests re-writing the introduction section in order to clarify the research question and the objectives. He also recommends developing the theoretical background of the stable isotope method and adding more references. Finally, he proposes to develop

the Materials and Methods section by moving all the section 5.1 in this section, and to clarify our use of Lake Ihotry from Madagascar as a test of our method. We understand how several paragraphs can be considered as misplaced, thus precluding a good understanding of our study. We will be ready to work on improving the text following his recommendations.

We agree with the reviewer that several previous studies were able to determine the hydrological budget of tropical lakes, even when difficult to reach, and that some of them also used satellite altimetry data. However, all these studies dealt with relatively small basins, and where based in general on extensive data sets and continuous monitoring. For instance, Liebe et al. (2005) and Rodrigues et al. (2012) (cited in the paper), estimated small reservoir storage capacity with altimetry data. Gal et al. (2016) calculated a hydrological budget with altimetry data, but with regular water height measurements (data we do not have in our study). Alternative methods such as the Thornthwaite-Mather procedure (Steenhuis and Van der Molen, 1986; Collick et al., 2009) are also efficient for estimating water budgets in remote areas, but require precise precipitation data, which remains unavailable in many areas. By contrast, we want to emphasize the huge surfaces of the catchments of the two lakes Iro and Fitri (195 000 and 96 000 km2), with only one meteorological station. It is thus impossible to interpolate data at this scale. Standard hydrogeological approaches are thus highly unconstrained in such a case, and isotopes may help to obtain a first order hydrologic budget. Calculating the mean evaporation/inflow ratio of a lake from a simple mass balance equation has been already applied in a number of studies (Gat, 1978; Gibson et al., 1993; Gibson, 2002; Yi et al., 2008). However, the seasonal variation of isotopic composition was generally not discussed in studies despite very few available data over a year (Mayr et al., 2007; Yuan et al., 2011; Brooks et al., 2014). This may induce large error on the results, especially in catchments with seasonal climate contrast (Gibson, 2002).

In our manuscript, we describe a simple method to circumvent this difficulty, by combining isotopes with altimetry data in order to assess uncertainties on the influx and

outflux calculation. This approach requires only point data collected during the dry season, and is thus most appropriate for isolated tropical lakes with difficult access during the wet season. Recently, Cui et al. (2018) reviewed a number of case-studies where detailed hydrographic and isotopic monitoring were obtained on different lakes under various climates, in order to identify the most representative period for sampling during the seasonal cycle in each case, in the hope of obtaining a representative long term perspective on the lake water balance. We see our study as fully complementary to this approach.

We also take good note that introducing lake Ihotry from Madagascar only in the discussion was too late, and thus confusing. We used our results previously published on this lake as a benchmark test of our method, and did not want to place those results on the same line as the new data produced on the Chadian lakes during this study. However, the reviewer is right that this should be introduced and explained earlier in the paper.

We are willing to take into account all the reviewers recommendations to clarify our manuscript. We will re-write the introduction section in order to better put forward our research questions and objectives, and better develop the theoretical background of our use of the stable isotopes method. We will develop the Material and Method section to clarify our use of Lake Ihotry from Madagascar. In this section, we will also move all the section 5.1 to explain our scientific approach, in order to facilitate understanding of the discussion section.

Brooks, J. R., Gibson, J. J., Birks, S. J., Weber, M. H., Rodecap, K. D. and Stoddard, J. L.: Stable isotope estimates of evaporation : inflow and water residence time for lakes across the United States as a tool for national lake water quality assessments, Limnology and Oceanography, 59(6), 2150–2165, doi:10.4319/lo.2014.59.6.2150, 2014.

Collick, A. S., Easton, Z. M., Ashagrie, T., Biruk, B., Tilahun, S., Adgo, E., Awulachew, S. B., Zeleke, G. and Steenhuis, T. S.: A simple semi-distributed water balance model for the Ethiopian highlands, Hydrological Processes, (23), 3718–3727, doi:10.1002/hyp.7517, 2009.

Cui, J., Tian, L. and Gibson, J. J.: When to conduct an isotopic survey for lake water balance evaluation in highly seasonal climates, Hydrological Processes, 32(3), 379–387, doi:10.1002/hyp.11420, 2018.

Gal, L., Grippa, M., Hiernaux, P., Peugeot, C., Mougin, E. and Kergoat, L.: Changes in lakes water volume and runoff over ungauged Sahelian watersheds, Journal of Hydrology, 540, 1176–1188, doi:10.1016/j.jhydrol.2016.07.035, 2016.

Gat, J. R.: Isotope hydrology of inland sabkhas in the Bardawil area, Sinai, Limnology and Oceanography, 23(5), 841–850, 1978.

Gibson, J. J.: Short-term evaporation and water budget comparisons in shallow Arctic lakes using non-steady isotope mass balance, Journal of Hydrology, 264(1), 242–261, 2002.

Gibson, J. J., Edwards, T. W. D., Bursey, G. G. and Prowse, T. D.: Estimating Evaporation Using Stable Isotopes: Quantitative Results and Sensitivity Analysis for Two Catchments in Northern Canada, Hydrology Research, 24(2–3), 79–94, 1993.

Liebe, J., Van de Giesen, N. and Andreini, M.: Estimation of small reservoir storage capacities in a semi-arid environment, Physics and Chemistry of the Earth, Parts A/B/C, 30(6), 448–454, doi:10.1016/j.pce.2005.06.011, 2005.

Mayr, C., Lücke, A., Stichler, W., Trimborn, P., Ercolano, B., Oliva, G., Ohlendorf, C., Soto, J., Fey, M., Haberzettl, T., Janssen, S., Schäbitz, F., Schleser, G. H., Wille, M. and Zolitschka, B.: Precipitation origin and evaporation of lakes in semi-arid Patagonia (Argentina) inferred from stable isotopes ($\delta$18O, $\delta$2H), Journal of Hydrology, 334(1–2), 53–63, doi:10.1016/j.jhydrol.2006.09.025, 2007.

Rodrigues, L. N., Sano, E. E., Steenhuis, T. S. and Passo, D. P.: Estimation of Small

Reservoir Storage Capacities with Remote Sensing in the Brazilian Savannah Region, Water Resources Management, 26(4), 873–882, doi:10.1007/s11269-011-9941-8, 2012.

Steenhuis, T. S. and Van der Molen, W. H.: The Thornthwaite-Mather procedure as a simple engineering method to predict recharge, Journal of Hydrology, 84(3–4), 221–229, 1986.

Yi, Y., Brock, B. E., Falcone, M. D., Wolfe, B. B. and Edwards, T. W. D.: A coupled isotope tracer method to characterize input water to lakes, Journal of Hydrology, 350(1–2), 1–13, doi:10.1016/j.jhydrol.2007.11.008, 2008.

Yuan, F., Sheng, Y., Yao, T., Fan, C., Li, J., Zhao, H. and Lei, Y.: Evaporative enrichment of oxygen-18 and deuterium in lake waters on the Tibetan Plateau, Journal of Paleolimnology, 46(2), 291–307, doi:10.1007/s10933-011-9540-y, 2011.

---

## Author Response (AR1)

Dear Editor,

Please find enclosed the revised version of our manuscript entitled "Unraveling the hydrological budget of isolated and seasonally contrasted sub-tropical lakes" (Poulin et al., 2018), modified following your indications received on 27/08/2018.

As you pointed out, both reviewers considered that our paper was suffering from a lack of clarity. The first reviewer missed the point why we used stable isotopes, and what objectives we followed. The second reviewer also found the structure of the manuscript difficult to follow and a lack of clearly stated objectives. He suggested to rewrite the introduction, and to re-arrange the Materials and Methods section in order to clarify our use of Lake Ihotry from Madagascar as a test of our method.

These detailed and constructive recommendations proved very useful for us in re-organizing the manuscript following a more classical structure:

– We extended the Introduction section in order to clarify the situation of our study in the more general context of the hydrological studies of remote inter-tropical lakes. We also developed more explicitly the stable isotope background, in order to better explain our specific contribution of using remote sensing data to exploit a single point isotopic data in the seasonal variation. Our objectives are now explicitly listed at the end of the Introduction section.

– We moved the description of lake Ihotry towards the section describing the study sites.

– We created a new paragraph about the data collection, distinct from the Method section.

– We moved the previous sections 5.1 and part of 5.2 (formerly in the discussion) into the new Methods section. This section is now divided in two parts: the first one explains our hydrologic budget calculation at the lake scale, and the second one our discussion at the watershed scale.

– We tried to improve the clarity of part 5.2 (now 6.2) about partitioning transpiration from evaporation in watershed studies.

The English language has been edited by a professional reader.

We hope that these changes will have improved the understanding of our work, and we thank again the reviewers and yourself for your time on this manuscript.

Two copies of the revised manuscript are included, the final revised version and a marked copy. The paragraphs that have been added or modified are highlighted in red and the paragraphs that have been moved are in blue.

We thank you in advance for your consideration and look forward to hearing from you.

Yours sincerely,

[revised manuscript text omitted]